# Design and Optimization of Lornoxicam Dispersible Tablets Using Quality by Design (QbD) Approach

**DOI:** 10.3390/ph15121463

**Published:** 2022-11-25

**Authors:** Nawaf Almotairi, Gamal M. Mahrous, Saleh Al-suwayeh, Mohsin Kazi

**Affiliations:** Department of Pharmaceutics, College of Pharmacy, King Saud University, P.O. Box 2457, Riyadh 11451, Saudi Arabia

**Keywords:** lornoxicam, dispersible tablets, quality by design, dissolution improvement, stability

## Abstract

The present study aims to design and optimize the lornoxicam dispersible tablet (LXDT) formulation using the Quality by design (QbD) approach. A randomized Box–Behnken experimental design was used to characterize the effect of the critical factors, such as filler (MCC/Mannitol) ratio, mixing time, and disintegrant concentration, and assessed for their impacts on the critical quality attributes (responses), including dispersibility time, friability, dissolution efficiency, and content uniformity, respectively. The drug-excipients interaction of the formulation was investigated using FTIR and DSC, respectively. The accelerated stability study at 40 °C/75% relative humidity was performed. FTIR revealed an absence of any significant chemical interaction in solid state. DSC thermogram suggested that LX endothermic peak was slightly decreased due to the dilution effect. LXDT formulations exhibited acceptable friability (0.2 to 0.9%). The dissolution efficiency of LXDT formulations ranged from 72.21 to 93.63%. The overall study showed that the optimum level of independent factors was found to be 3:1 MCC/Mannitol, 11 min mixing time, and 6.23% disintegrant concentration. Accelerated stability studies showed the compendial acceptable hardness, friability, and disintegration times. The application of QbD approach can help in the detailed understanding of the effect of CMAs and CPPs on the CQAs on LXDT final product.

## 1. Introduction

Dispersible tablets (DT) can be prepared by many different techniques like spray drying, lyophilization, sublimation, molding, cotton candy processing, and direct compression (DC). The basic approach to develop DTs is the use of superdisintegrants, such as crospovidone, croscarmellose sodium, and sodium starch glycolate [1]. Other approaches include freeze drying and vacuum drying techniques to maximize the pore structure of the tablet matrix [2]. However, freeze drying is unsuccessful due to the fragile and hygroscopic product yields [3]. Vacuum drying, along with the sublimation of volatilizable ingredient, has been employed to increase tablet porosity. The main characteristic of DTs is their ability to disintegrate or dissolve rapidly in a liquid and have a pleasant mouth feel [4]. Several scientists reported different methods which were adopted for the manufacturing of DT. Samprasita et al. developed orally disintegrating tablets by the ion exchange resin method [5]. Kulkarni et al. developed fast disintegrating oral meloxicam tablets by the wet granulation method [6]. El-Mahrouk et al. formulated orally dispersible meloxicam capsules using beta-cyclodextrin [7]. Rangasamy et al., carried out Β-Cyclodextrin complexation of meloxicam using the Kneading method followed by direct compression method to achieve rapid disintegration of tablet [8]. Inamdar et al. prepared solid dispersions of meloxicam using different polymers to facilitate the dissolution profiles of the tablets [9]. Dehghana and Jafar developed solid dispersions of meloxicam, and their objective was to compare several methods, i.e., physical mixing, co-grinding technique, and solvent evaporation procedure, to increase the release pattern of meloxicam [10].

DC is a popular choice because it provides the shortest, most effective, and least complex way to produce tablets. The manufacturer can blend an API with the excipient and the lubricant, followed by compression, which makes the product easy to process. No additional processing steps are required. Moisture or heat-sensitive ingredients, which would be contraindicated in wet granulation, can also be used in this type of process. However, it does require a very critical selection of excipients in comparison to granulation processes because the raw materials must demonstrate good flowability and compressibility for successful operation [11].

Lornoxicam (LX), is a nonsteroidal anti-inflammatory drug (NSAID) that belongs to the oxicam class. LX has analgesic, anti-inflammatory, and antipyretic properties, and differs from other oxicam compounds due to its potent inhibition of prostaglandin biosynthesis. The development of LX as dispersible tablet is expected to provide quick dissolution and rapid absorption, which may produce rapid onset of action with minimal gastric irritation. The aim of the present study is to design and optimize lornoxicam dispersible tablet (LXDT) formulation using the Quality by design approach (QbD). QbD encompasses designing and optimizing formulations and manufacturing processes that ensures meeting predefined product specifications. The main purpose is to switch from the quality by testing procedure to a development, which could improve the understanding of processes and products, hence improving product quality, process efficiency, and regulatory flexibility.

## 2. Results and Discussion

### 2.1. Particle Size Analysis

The particle size of the active agent plays a key role in the physical stability and bioavailability of the drug product. The rates of sedimentation and agglomeration are affected by the particle size. If the particle size is reduced to half of its original size, the rate of sedimentation decreases by a factor of four. In addition, a drug with small particle size is expected to increase the surface area, which is expected to result in increasing the drug dissolution rate and vice versa [12]. In the current investigation, the average particle size of the representative LX-DT provided size range between 1 to 10 μm with the mean particle size of 4.25 μm and span value of 0.8 along with a good distribution (Figure 1).

### 2.2. Differential Scanning Calorimetry (DSC)

DSC analyses was used to predict any possible interaction between LX and the formulation ingredients. Figure 2 shows differential scanning calorimetry (DSC) scans of LX alone and with excipients in their physical mixtures. DSC thermogram of LX alone has an endothermic peak at 225 °C, which corresponds to the melting point of LX indicating the crystallinity of the drug. The small endothermic peak that appeared before the drug melting peak may be due to decomposition [13], while Mannitol showed an endothermic peak 170 °C. The drug endothermic peak was slightly decreased due to the dilution effect but still existed in case of the physical mixture. The same behavior was observed with the other excipients indicating no physical interaction between LX and the used excipients.

### 2.3. Fourier Transform Infrared Spectroscopy (FTIR)

The FTIR spectra was recorded for LX and powder mixtures of the formulation to assess any possible chemical interactions between LX and the excipients. Figure 3 demonstrates the FTIR spectra of the untreated Lornoxicam, CP, MCC, Mannitol, and their formulations. The FTIR of LX (pure drug) shown intense bands at 3060 cm^−1^, 1647 cm^−1^, 1590 cm^−1^, 1323 cm^−1^, and 783 cm^−1^ corresponding to the functional groups NH, C=O, CONH-, SO_2_, and C-Cl bending, respectively [14]. CP presented a characteristic peak at 2976 cm^−1^ (C-H stretching vibration), the peak at 1622 cm^−1^ was due to the C=O stretching and 1277 cm^−1^ (C-N stretching) [15]. MCC presented a characteristic peak at ~3324.1 cm^−1^ (N-H stretching vibration) and 1022 cm^−1^ (C-O stretching vibration) [16]. Mannitol presented a characteristic peak at ~3281 cm^−1^ (-OH stretching vibration), ~2947 cm^−1^ (C-H stretching vibration), and 1075 cm^−1^ (C-O stretching vibration) [17]. The stretching vibration of all the above-mentioned functional groups were found to be within range in all the prepared LXDT indicating absence of any significant chemical interaction in solid state.

### 2.4. Flow Properties of Powder Blends

In order to specify the flow characteristics of LXDT powder blends, the angle of repose and tapped volume was measured (Table 1). The results show that LXDT has passable flow properties to be compressed directly by tablet machine with the angle of repose range of (41–44°) and Carr’s index range of (21–23%).

### 2.5. Evaluation of LX Dispersible Tablets Properties

Tablet uniformity is important and is used to make sure that every tablet contains the amount of drug substances intended with only little variation among tablets within a batch [18].

Tablets require a certain amount of strength, or hardness, to withstand mechanical shocks of handling in manufacturing, packaging, and shipping. Recently, the relationship of hardness to tablet disintegration and the drug dissolution (release) rate has become apparent. Tablet hardness has been defined as the force required to break a tablet in a diametric compression test.

The thickness of the tablet is the only dimensional variable related to the compression process. Tablet thickness is consistent batch to batch or within a batch only when the tablet granulation or powder blend is adequately consistent in particle size and size distribution, the punch tooling is of consistent length, and the tablet press is clean and in good working order. Thickness should be controlled within ±5% variation of a standard value. Thickness must be controlled for consumer acceptance of the product and to facilitate packaging [19].

Tablet friability is an important characteristic that measures the resistance of the tablets to shipping and abrasion. The purpose of having friability test is to make sure that the formed tablets are able to withstand mechanical stresses during their manufacturing, distribution and handling by the end-user [20].

The evaluated properties of LXDT were shown in Table 2 The drug content of the prepared LXDT was found to be within the pharmacopoeial guidelines (USP 41-NF36) requirement, the acceptant value of content uniformity in all formulations ranged from 3.15 to 14.82%. LXDTs exhibited a thickness from 3.15 mm to 3.574 mm and Hardness from 3.30 ± 0.74 to 4.84 ± 1.09 Kp, which indicated that tablets had good mechanical resistance. LXDTs friability were ranged from 0.20% to 1.0%, which are in an acceptable range; NMT 1%. Dispersible tablets are required to disintegrate within 3 min in water, LXDTs exhibited a dispersibility time from 6 s to 20 s. The swallowing of dispersion can prevent localization of the drug in the stomach and hence decrease gastric irritation. 

### 2.6. In Vitro Dissolution Studies

The release profiles for the fifteen formulations are presented in Figure 4. The cumulative percent released after 60 min of LXDT formulations ranging from 84.65% in (LXDT F12) to 103.79% in (LXDT F11). The results illustrated that there is a relationship between the disintegrant concentration and dissolution profile. Formulation with low disintegrant concentration (2%) LXDT F7, LXDT F12, and LXDT F14 showed the lowest cumulative amount of LX release (85%, 84.6%, and 88%, respectively). 

In contrast to the formulation with the disintegrant concentration (6%) LXDT F2, LXDT F4, and LXDT F11 showed the highest cumulative amount LX release (99.23%, 97.18%, and 103.79%, respectively). This result might be attributed to the short disintegration time of LX lead to rapid breakdown of the tablet into small particles thus increase the surface area exposing to the medium and enhancing the dissolution of the drug and vice versa [21].

### 2.7. Response Surface Methodology for Optimization of LXDT

The Box–Behnken design was utilized for optimization of LXDTs with a minimum dispersibility time and with acceptable friability, content uniformity, and with maximum dissolution efficiency. The experimental design matrix with different levels of independent factors is compiled in Table 3. The fifteen runs of the experiment were assessed for the dispersibility time (Y_1_), tablet friability (Y_2_), dissolution efficiency (Y_3_), and content uniformity (Y_4_) (Table 3).

### 2.8. Statistical Analysis and Summary of Fit 

A different statistical model, expressing linear, quadratic, interactive, and polynomial terms, was applied to evaluate the influence of the control factors on the responses. Table 4 summarizes the coefficients of model terms and associated *p* values for Y1–Y4. If the *p* value was less than 0.05 (*p* < 0.05), the factor could be considered to affect the responses (Y1–Y4) significantly. For simplicity of the regression model, the non-significant terms (*p* > 0.05) were not considered.

A positive or negative coefficient indicated an increase or decrease of the corresponding response, respectively. Analysis of variance (ANOVA), R^2^, adjusted R^2^, and predicted R^2^ were determined to validate the experimental design (Table 4). The high values of R^2^, adjusted R^2^, and predicted R^2^ showed well-fitted responses. In addition, *p* values of lack of fit of Y1–Y4 were 0.2985, 0.1542, 0.44, and 0.3512, respectively, which were greater than 0.05 for all responses, suggesting the insignificant model errors. 

### 2.9. Influence of Independent Variables on the In-Vitro Dispersibility Time (Y1)

The dispersibility time is an important parameter, which performs a significant role in the release pattern of DTs and, consequently, absorption via the biological membranes. Fast disintegration of DTs is highly desirable to ensure tablets rapid break down into smaller fragments so that the largest surface area can be created for faster dissolution of LX [22]. LXDT formulations showed marked variations in dispersibility times ranging from 6–20 s (Table 2). Due to the highest correlation coefficient (R^2^) values and the lowest predicted residual error sum of squares (PRESS), the data of dispersibility was fitted to the quadratic model. In addition, the predicted and adjusted R^2^ indicating reasonably the validity of the model. Furthermore, adequate precision for response was greater than 4, indicating an adequate signal-to-noise ratio, which implies the suitability of the selected model to explore the design space.

ANOVA for the dispersibility time confirmed the significance of the quadratic model as depicted by its F-value of 53.36 (*p* = 0.0001). The lack of fit *p*-value of 0.2981 (*p* >0.05) implied a non-significant lack of fit relative to pure error, warranting data fitting to the proposed model. The representative equation for quadratic model was generated in terms of coded factor as follows:Y1 = 7.33 − 2.75 X_1_ + 0.25 X_2_ − 3.5 X_3_ − 0.25 X_1_X_2_ − 0.25 X_1_X_3_ + 0.25 X_2_X_3_ + 0.7083 X_1_^2^ + 1.21 X_2_^2^ + 6.21 X_3_^2^(1)

The statistical analysis revealed that the filler ratio (X_1_) and the superdisintegrant concentration (X_3_) have a significant negative effect on dispersibility time (*p* < 0.0002 and *p* < 0.0001, respectively). The quadratic terms corresponding to these investigated variables, in addition to the interaction term X_1_X_3_ corresponding to the interaction between filler ratio, and the disintegrant concentration were found to be insignificant.

Figure 5 illustrates the influence of independent variables and their interaction on the dispersibility time the response surface plots for the effects of the studied variables on dispersibility time. It was evident that the dispersibility time significantly decreases with an increase in filler ratio. This is expected as MCC is used usually to enhance disintegration. In addition, increasing the disintegrant up to 6% resulted in a decrease of dispersibility time. Generally, crospovidone is used at the concentration of 2–5% *w/w* [23]. Crospovidone levels higher than 8% of tablet weight produces weaker tablets with a low disintegration rate [24]. In this study, increasing disintegrant concentration up to 6% leads to significantly decreases of dispersibility time, and, by increasing disintegrant concentration from 6–10%, the dispersibility time was increased.

### 2.10. Influence of Independent Variables on the Tablet Friability (Y2)

The friability test tells how much mechanical stress tablets are able to withstand during their manufacturing, distribution, and handling by the customer [25]. All the DT formulations exhibited acceptable friability, which was less than 1%, and it is ranged from 0.2% to 1.0% of different experimental runs as shown in Table 2. 

Similar to the dispersibility time, the data of friability was also fitted to the linear model (Table 4) based on the highest correlation coefficient (R^2^) and the lowest predicted residual error sum of squares (PRESS). In addition, their predicted and adjusted R^2^ values were rational enough to signify the validity of the model. Furthermore, adequate precision for a response was greater than 4, indicating an adequate signal-to-noise ratio, which implies the suitability of the selected model to explore the design space. ANOVA for the friability confirmed the significance of the linear model as depicted by its F-value of 37.95 (*p* < 0.0001). The lack of fit *p*-value of 0.154 implied a non-significant lack of fit relative to pure error, ensuring data fitting to the proposed model. The equation representing the leaner model was generated in terms of coded factor as follows:Y2 = 0.5613 − 0.29 X_1_ − 0.0087 X_2_ + 0.088 X_3_(2)*p* value < 0.0001 represents a significant effect of the corresponding factors on tablet friability. The coefficients of X_3_ were positive, while the coefficients of X_1_ and X_2_ were negative, suggesting that tablet friability decreased with the increase of filler ratio and mixing time, and increased with the increase of disintegrant concentration. The actual model R^2^, adjusted R^2^, and R^2^ predicted, for tablet friability (Y2) were 0.9119, 0.8879, and 0.8242, respectively. The similarity of these values was suggestive of the goodness of fit.

Figure 6 displays the effects of filler ratio, mixing time, and disintegrating concentration, and their combined interaction on tablet friability. As the filler ratio (X_1_) decreased, tablet friability significantly increased (*p* < 0.0001). Tablet friability decreased as mixing time (X_2_) increased from 5 to 15 min, However, this effect was not significant (*p* = 0.7640), also, as the disintegrant concentration (X_3_) increased from 2 to 10%, tablet friability significantly increased (*p* = 0.0097). 

### 2.11. Influence of Independent Variables on the Dissolution Efficiency (Y3)

Drug release is a major step in dissolution process which determines oral bioavailability of the model drug. The drug must be released from the dosage form and remain solubilized in the gastrointestinal tract to have maximum absorption. Thus, drug dissolution rate is a consideration of great property for LXDT formulations, Therapeutic agents with poor aqueous solubility (less than 100 μg/mL) often present dissolution limitations to absorption. LX is a BCS (biopharmaceutical classification system) class II compound with poor aqueous solubility [20].

Dissolution efficiency (DE) of LXDT was calculated in the studies according to the following equation: (3)$$\mathrm{DE}=\frac{\int_{t_{1}}^{t_{2}}y\;\cdot \;dt}{y_{100}\times \left(\;t_{2}-t_{1}\right)}\times 100$$
where *y* represents the percent of dissolved product, DE is the area under the dissolution curve between time points t_1_ and t_2_, expressed as a percent of the curve at maximum dissolution, and *y*_100_, over the same time, respectively.

The DE% after 30 min of LXDT formulations ranged from 64.00% in (LXDT F7) to 79.30% in (LXDT F2) (Figure 6). Based on the highest correlation coefficient (R^2^) and the lowest predicted residual error sum of squares (PRESS), the data of Dissolution efficiency (Y3) fitted to the quadratic model. In addition, there was a reasonable agreement between the predicted and adjusted R^2^ indicating the validity of the model. Furthermore, adequate precision for response was greater than 4, indicating an adequate signal-to-noise ratio, which implies the suitability of the selected model to explore the design space.

ANOVA for the dissolution efficiency confirmed the significance of the quadratic model as depicted by its F-value of 9.23 (*p* = 0.0124). The lack of fit *p*-value of 0.33 implied a non-significant lack of fit relative to pure error, ensuring data fitting to the proposed model. 

The actual model R^2^, adjusted R^2^, and R^2^ predicted value for Y3 were 0.9432, 0.8410, and 0.651, respectively, The Predicted R^2^ is in reasonable agreement with the Adjusted R^2^. 

The model proposes the following polynomial equation for dissolution efficiency:Y3 = 76.47 + 1.26 X_1_ + 0.0875 X_2_+ 4.33 X −0.8750X_1_X_2_ + 1.05 X_1_X_3_ − 1.05 X_2_ X_3_ + 0.7792 X_1_^2^ + 1.03X_2_^2^ − 6.50 X_3_^2^(4)

The obtained *p* value 0.0124 represents a significant effect of the corresponding factors on dissolution efficiency. In this case, X_3_ and X_3_^2^ are significant model terms. 

The effects of filler ratio, mixing time, and disintegrant concentration on dissolution efficiency are shown in Figure 7 and Figure 8, and Table 3. As the filler ratio (X_1_) increased, the dissolution efficiency increased (*p*= 0.1397). The dissolution efficiency increased as mixing time (X_2_) increased from 5 to 15 min. However, this effect was not significant (*p* = 0.9079). Additionally, the dissolution efficiency significantly increased (*p* = 0.0018) as the disintegrant concentration (X_3_) increased from 2% to 6% and then decreased by increasing the concentration to 10%. These results are in accordance with dispersibility results.

### 2.12. Influence of Independent Variables on the Content Uniformity (Y4)

The acceptance values (AV) were calculated to determine the tablet content uniformity. The AV of the experimental formulations ranged from 3.152% to 14.82% (Table 3). Based on the highest correlation coefficient (R^2^) and the lowest predicted residual error sum of squares (PRESS), the data of content uniformity (Y4) fitted to the quadratic model. The actual model R^2^, adjusted R^2^, and R^2^ predicted for Content uniformity (Y4) were 0.9586, 0.8841, and 0.4805, respectively. Furthermore, adequate precision for response was greater than 4 indicating an adequate signal-to-noise

The reduced regression equation in coded terms for Y4 is shown in Equation (5).
Y4 = 3.88 + 0.0036 X_1_ − 1.64 X_2_ + 0.7266 X_3_ − 0.5070 X_1_X_2_ + 0.4563 X_1_X_3_ − 0.539 X_2_X_3_ + 0.5954 X_1_^2^ + 6.36 X_2_^2^ + 0.7099 X_3_^2^(5)

The obtained *p* value < 0.05 of any of the factors represented a significant effect of the corresponding factors on the content uniformity. Mixing time showed the most significant effect on the content uniformity among the studied variables (*p* = 0.0134). The coefficients of X_2_, X_1_X_2_, and X_2_X_3_ were negative, while the coefficients of X_1_, X_3_, X_1_X_3_, X_1_^2^, X_2_^2^, and X_3_^2^ were positive. This suggested that the mixing time was inversely proportional to the AV, which means more uniformity in LX content (a low dose drug) whilst the filler ratio and disintegrant concentration were directly proportional to the AV.

The contour diagrams displaying the effect of MCC/ Mannitol ratio (X1), mixing time (X2), and disintegrant concentration (X3) on the content uniformity AV of LXDTs is shown in Figure 9. As the filler ratio (X_1_) increased, content uniformity AV increased. In addition, content uniformity AV increased as disintegrant concentration (X_3_) increased. However, these effects were not significant (*p* = 0.9937 and 0.1586, respectively). Content uniformity AV significantly increased (*p* = 0.0134) as the mixing time (X_2_) decreased. 

### 2.13. Optimization 

Numerical optimization was conducted by minimizing dispersibility time (Y_1_), friability (Y_2_), and the acceptance value of content uniformity (Y_4_) responses while maximizing the drug dissolution (Y_3_) using the Expert Design 12 program. The optimum level of independent factors was found to be 3:1 MCC/Mannitol, 11-min mixing time, and 6.23% disintegrant concentration. The predicted and observed values of the responses for the optimized LX dispersible tablet formula are shown in Table 5. The prepared DTs showed comparable observed responses to the predicted ones in terms of dispersibility time, friability, dissolution efficiency, and content uniformity, ensuring the validity and predictability of the employed experimental design.

### 2.14. The Stability of LXDT 

#### 2.14.1. Hardness 

The resistance of tablets to capping, abrasion, or breakage under conditions of storage, transportation, and handling before usage depends on its hardness. Tablet hardness is defined as the load required to crush or fracture a tablet placed on its edge. Sometime it is also termed tablet crushing strength [26]. In this study, ten tablets of each interval were subjected to the hardness test and the crushing strength of the tablet was measured. Average hardness of the tablets was calculated, and standard deviation was determined. Table 6 shows the hardness of 258 mg LX dispersible tablet with the hardness data around 4.00 ± 0.7.

#### 2.14.2. Friability 

The friability of the dispersible tablet containing LX nanoparticles showed 0.36% in initial time while it was 0.53% in the end of accelerated stability study. All results shown in Table 9 are complies with USP requirements [27]. In addition, a friability value less than 1% is compendially acceptable. All tablets were good looking and non-sticky. The color and shape of the tablets were observed visually. The thickness test of the tablets (Table 6) was performed on 20 tablets from each interval and the thickness test results were in accordance with the British Pharmacopeia for all samples.

#### 2.14.3. Dispersibility Time 

Dispersibility is a physical process related to the mechanical breakdown of a tablet or granulate particle into smaller particles when DT exposed to aqueous media [28], representing the breakage of inter-particle interactions generated during tablet compaction of granulated particles of the active pharmaceutical ingredient (API) and excipients. Upon aqueous dispersion, the tablet surface was wet and [29] the dispersibility time of the dispersible tablet containing LX was within 4 to 5 s, as shown in Table 6. 

#### 2.14.4. Drug Content during Stability Study 

The drug content was determined by estimating the API content in individual strip. The limit of content uniformity is 85–115% with the standard deviation of less than or equal to 6% according to USP [30]. Drug content determination is important so as to get accuracy in dosing [31]. The drug content of the prepared lornoxicam DT was found to be within the pharmacopoeial guidelines (USP41-NF36) requirement. The LXDT content in all stability intervals ranged from 95.14 ± 1.03 to 99.62 ± 0.77 (Table 6).

The in vitro dissolution profiles of LXDT, was studied after undergoing an accelerated stability study at 40 ± 2 °C/75% RH ± 5% RH for 6 months. Furthermore, it was compared with commercial (Xefo^®^ 8 mg). Table 7 shows the effect of storage at accelerated condition on the dissolution profile of LXDT. The dissolution data indicated a minor decrease in LX dissolution rate after 3 months and 6 months of storage compared to the rate at initial time, but it is not a significant change. In addition, the optimized formula dissolution (at initial time) was compared to the dissolution of LX from commercial tablets (Xefo^®^ 8 mg). 

The Similarity factor values (f2) calculated using the following equation:(6)$$f2=50\log\{\left[1+\frac{1}{n}\overset{n}{\underset{n=1}{\sum }}\left(R_{t}-T_{t}\right){\;}^{2}\right]{\;}^{-0.5}\times 100\}$$

According to the Food and Drug Administration’s (FDA) guidelines, similarity factor values (f2) greater than 50 (50–100) means similarity of the dissolution profiles (Diaz et al., 2016). It was observed that the f2 value was 95.8. From these results, it could be concluded that the LXDT showed a better in vitro dissolution profile (Table 7), so the dissolution was the higher as compared with commercial tablet (Xefo^®^ 8 mg).

## 3. Materials and Methods

### 3.1. Materials 

Lornoxicam (LX) was obtained from Tabuk Pharmaceuticals (Tabuk, Saudi Arabia). Microcrystalline cellulose (MCC) was obtained from Riedel-de Haën (Seelze, Germany). Crospovidone (CP) and Aspartame were kindly supplied by DPE pharma, GmbH & Co. KG (Germany). Sodium lauryl sulfate (SLS) was purchased from Avonchem (Macclesifeld, UK). Mannitol was obtained from Qualikems Fine Chem Pvt. Ltd. (Vadodara, India). All chemicals used in the current studies were of analytical grade.

### 3.2. Methods

#### 3.2.1. Particle Size Analysis

The particle size distribution of the dry LX powder was determined using laser light diffraction (Mastersizer Scirocco 2000, Malvern Instruments, UK). For a typical experiment, about 500 mg of LX was fed into the sample micro feeder. Samples were analyzed five times and average results were taken.

#### 3.2.2. Experimental Design

It is enviable to develop an acceptable pharmaceutical formulation using a minimum number of man-hours and raw materials in the shortest possible time. Pharmaceutical formulations are traditionally developed by changing one variable at a time, which is the tedious process to continue. However, it may be difficult to develop an ideal formulation using this classical technique. Therefore, it is very vital to understand the complexity of pharmaceutical formulations by using established statistical tools such as factorial design [32].

In this study, randomized factorial experimental design (Expert Design 11, Stat-ease Inc., Minneapolis, MN, USA) was employed using the Box–Behnken response surface methodology. It was applied to study the effect of critical factors on various quality attributes of LXDT (Table 8). 

The selected independent factors were MCC/Mannitol ratio (X1), Mixing time (X2), and superdisintegrant concentration (X3). These factors were evaluated for their impacts on the critical quality attributes (responses) of DT including dispersibility time (Y1), tablet friability (Y2), dissolution efficiency at 30 min (Y3), and content uniformity (Y4). DSC and FTIR were used to predict any possible interaction between LX and the selected formulation ingredients. Suitable fitting models were selected by comparing *p*-values and R^2^ values. Response surface charts were plotted using the derived equations and used as a tool to explain the effects of process parameters on the resulting tablet attributes. The effect of these formulations variables was investigated for the optimized formula. Table 1 lists the design variables with its coded values and actual values, and Table 9 provides the factorial design layout, i.e., all possible 15 combinations, respectively. Table 10 shows the composition of formulations LXDT according to the matrix of formulation in Table 9.

**Table 9 pharmaceuticals-15-01463-t009:** Matrix of formulations in Box–Behnken design for Lornoxicam DT.

Formulation	Mcc/Mannitol Ratio (X1)	Mixing Time (X2)	Disintegrant Conc. (X3)
F1	2	5	10
F2	2	10	6
F3	1	10	10
F4	1	5	6
F5	2	15	2
F6	3	15	6
F7	1	10	2
F8	2	10	6
F9	3	10	10
F10	2	15	10
F11	3	5	6
F12	3	10	2
F13	1	15	6
F14	2	5	2
F15	2	10	6

#### 3.2.3. Flow Properties of Powder Blends

Flow properties were determined according to USP 41<1174> for the powder blends by measuring the angle of repose using funnel method and compressibility index using graduated cylinder method [33].

##### Differential Scanning Calorimetry (DSC)

DSC analysis was used to predict any possible interaction between LX and the formulation ingredients. It was performed using DSC-8000 (Perkin Elmer, Waltham, MA, USA). The samples (3–5 mg) of LX alone and with excipients in their physical mixtures were hermetically sealed in aluminum pans and heated at a scanning rate of 10 °C min–1 over a temperature range of 30–300 °C under dry nitrogen flow (30 mL min^–1^).

##### Fourier Transform Infrared Spectroscopy (FTIR) Analysis

FTIR analysis (FTIR Spectrum BX from Perkin Elmer LLC., MA, USA) was performed to assess the complexation and chemical properties of powdered samples, particularly if any possible interaction was existing between LX and its excipients. An appropriate amount of pure LX powder, LX + Mannitol, and other solid formulations were equipped by compressing the powders for 5 min at 5 bars on a KBr press and the spectra were scanned on the wavenumber range of 400–4400 cm^−1^.

#### 3.2.4. Tablet Direct Compression

For each formula, the particular amount of LX, MCC, mannitol, and crospovidone were mixed in a turbula mixer (type S27, Erweka, Apparatebau, Germany) for the specified time as per design. This blend was further mixed with 2% SLS as the lubricant [34] in the turbula mixer for an additional 3 min. Then, the powder blends were compressed on the single die tablet press (Erweka, EKO, Langen, Germany), using flat rounded 10 mm sized punches. The tablet weight was set at 250 mg of the total weight, containing 8 mg of LX powder.

#### 3.2.5. Evaluation of Tablets 

##### Hardness, Friability and Thickness

The hardness was determined using the hardness tester (Erweka TBH 28, Heusenstamm, Germany), assessing 10 tablets from each batch. The average hardness and standard deviation were calculated. Tablet friability was determined according to USP30-NF25. In short, 20 tablets were weighed (W1) and placed into the friabilator (Erweka, TA3R, Heusenstamm, Germany) with 25 rpm rotation for 4 min. The tablets were then taken out and reweighed after removal of fines (W2) to calculate the friability by the following equation:% Friability = 100 × (W1 − W2)/W1(7)

Ten tablets were randomly selected and individually measured for their thickness using a micrometer, and the average value ± SD was reported.

##### Weight Uniformity

The weight variation test was carried out according to the BP to ensure uniformity in the weight of the prepared tablets [35]. For each formulation batch, twenty tablets were randomly selected and accurately weighed individually, using a digital balance; their average weight were calculated and reported as mean ± SD. 

##### Content Uniformity

Content uniformity was evaluated according to the United States Pharmacopoeia guidelines [33]. Ten LXDT, weighing approximately 250 mg each (theoretically equivalent to 8 mg of LX), were accurately weighed, finely powdered, and then transferred individually into a volumetric flask. Approximately 60 mL of methanol was added, followed by ten drops of 1 M sodium hydroxide solution to ensure complete dissolution of the drug, and the volume were made up to 100 mL with methanol. The mixtures were shaken by mechanical means for 5 min, followed by sonication for 10 min. The dispersion was filtered, drug content was determined spectrophotometrically (type PS-303UV, Apel, Japan) at 378 nm using a constructed and validated calibration curve, and drug contents were calculated and reported. The acceptance value was calculated according to USP using the following equation: AV = [M − x^−^] + ks(8)
where AV is the acceptance value, K is the acceptability constant, and s is the standard deviation.

##### Dispersibility

Dispersibility was assessed by dispersing one tablet in 15 mL of distilled water, maintained at 25 ± 2 °C, using a 100-mL glass beaker. The water was stirred using a magnetic stirrer (50 rpm) until the tablet disperse completely. The time for the complete dispersion of the tablets were determined, and the mean of the dispersion time for six tablets were calculated [36].

#### 3.2.6. In Vitro Dissolution Studies

In vitro dissolution studies were conducted according to the USP guidelines using the USP dissolution apparatus II (Model 85T, Caleva Ltd., PA, USA). The paddles were stirred at 50 rpm [33]. The dissolution tests were performed using 800 mL of phosphate buffer (pH 6.8) as the dissolution medium and was maintained at 37 ± 0.5 °C. At predetermined time intervals (5, 10, 20, 30, and 60 min), a 5 mL sample was withdrawn through Millipore filter. The withdrawn samples were directly analyzed spectrophotometrically (type PS-303UV, Apel, Japan) at 378 nm, and the percent of LX released were determined as a function of time.

#### 3.2.7. Accelerated Stability Study

The optimized DT formulation was stored in a stability cabinet under storage conditions maintained, according to the ICH guidelines, at 40 ± 2 °C and RH of 75 ± 5%. At predetermined time intervals (0, 1, 3, and 6 months), samples were withdrawn and allowed to reach room temperature. The in vitro dispersion time, dissolution, hardness, friability, and content uniformity were tested for the withdrawn tablets. The drug content and dissolution testing were conducted using ultra performance liquid chromatography (UPLC) [37,38]. The physical appearance of the samples was finally examined to record any changes.

#### 3.2.8. Statistical Analysis

Expert Design 11 software was used to analyze the data. The data were expressed as mean ± standard deviation (SD). The significance was determined by applying one-way ANOVA. A value of *p* < 0.05 is denoted as significant throughout the current study.

## 4. Conclusions

LXDT optimized formula resulted in a decrease in dispersion time, acceptable hardness, friability, showed improved in vitro dissolution profile with acceptable taste, and good stability, and expected less gastric irritation as compared to the commercial tablet. Many factors can be controlled in order to achieve optimization results including formulation and process parameters. The application of QbD approach in the formulation of LXDT can help formulators in a detailed understanding of the effect of CMAs and CPPs on the CQAs of the final product.

## Figures and Tables

**Figure 1 pharmaceuticals-15-01463-f001:**
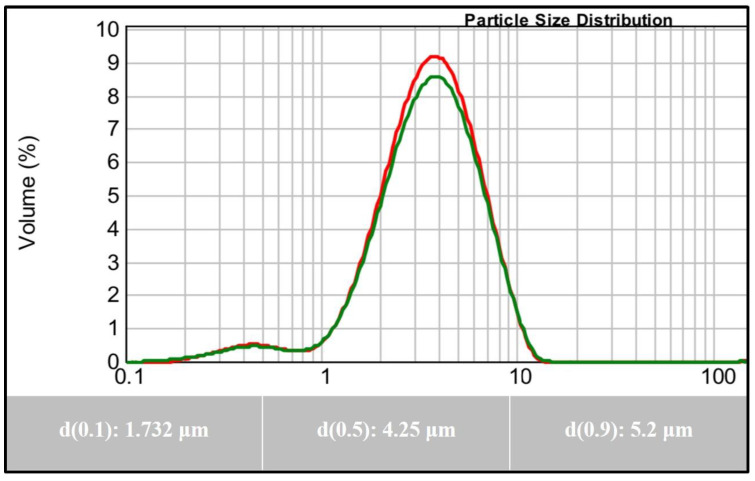
Particle Size Analysis of Lornoxicam.

**Figure 2 pharmaceuticals-15-01463-f002:**
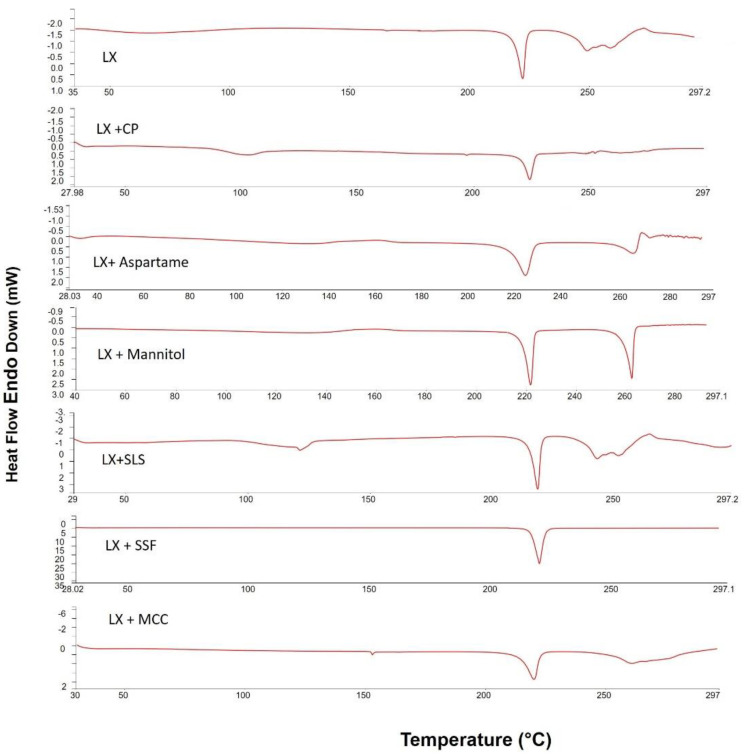
Differential scanning calorimetric thermograms of pure LX; LX + CP; LX + Manitol; LX + MCC; LX + SLS; LX + Aspartame; and LX + SSF.

**Figure 3 pharmaceuticals-15-01463-f003:**
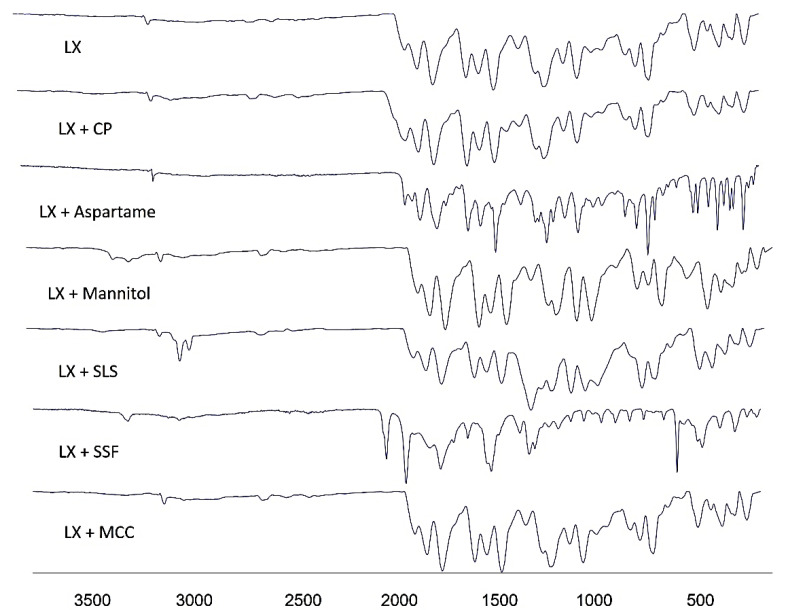
FTIR spectra of pure LX; LX + CP; LX + Manitol; LX + MCC; LX + SLS; LX + Aspartame; and LX + SSF.

**Figure 4 pharmaceuticals-15-01463-f004:**
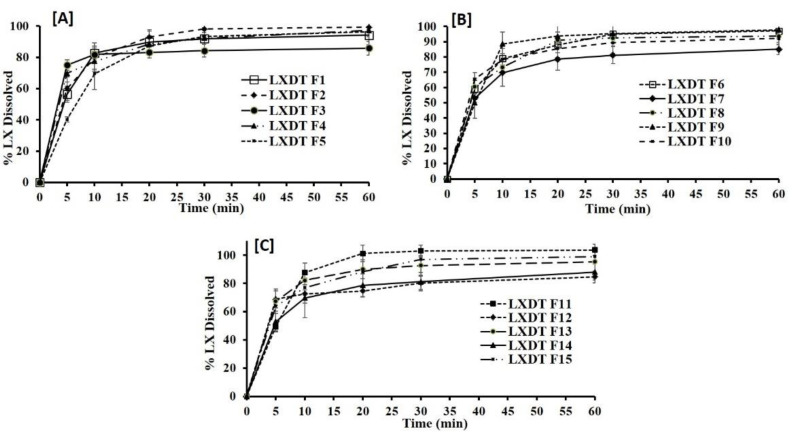
In vitro dissolution profile of LXDT formulations (**A**) LXTD F1–F5, (**B**) LXTD F6–F10, and (**C**) LXTD F11–F15.

**Figure 5 pharmaceuticals-15-01463-f005:**
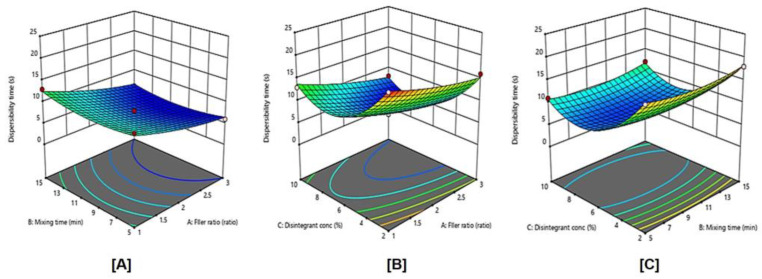
Influence of independent variables on the dispersibility time (Y1) with significance (**A**) *p* value = 0.4150, (**B**) *p* value = 0.0002, and (**C**) *p* value < 0.0001.

**Figure 6 pharmaceuticals-15-01463-f006:**
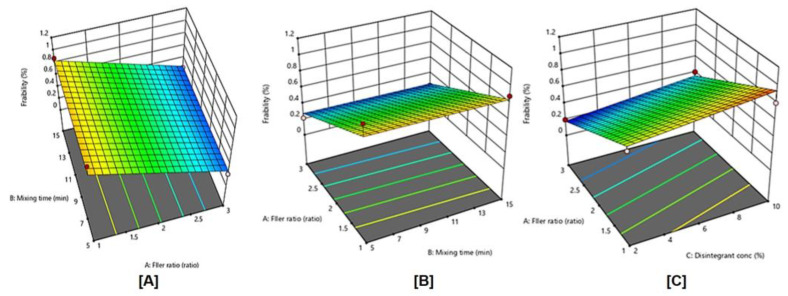
Influence of independent variables on tablet friability (Y2) with significance (**A**) *p* value = < 0.0001, (**B**) *p* value = 0.7640, and (**C**) *p* value < 0.0097.

**Figure 7 pharmaceuticals-15-01463-f007:**
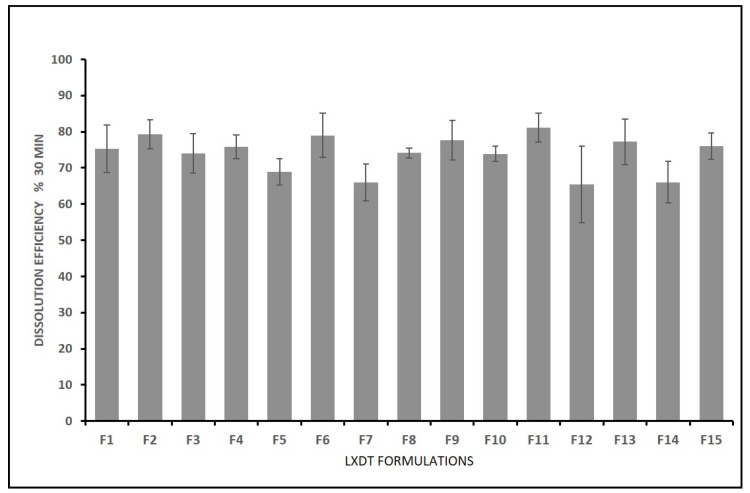
In vitro dissolution efficiency of LXDT after 30 min.

**Figure 8 pharmaceuticals-15-01463-f008:**
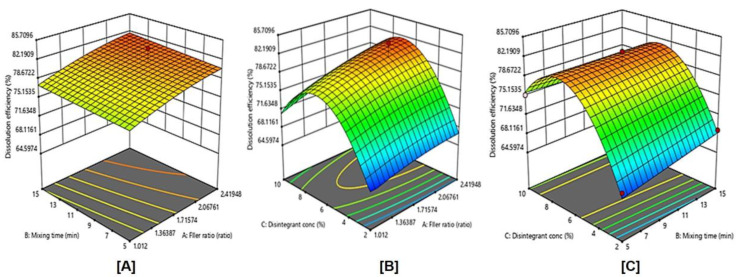
Influence of independent variables on the Dissolution efficiency (Y3) with significance (**A**) *p* value = 0.1397, (**B**) *p* value = 0.0018, and (**C**) *p* value < 0.9079.

**Figure 9 pharmaceuticals-15-01463-f009:**
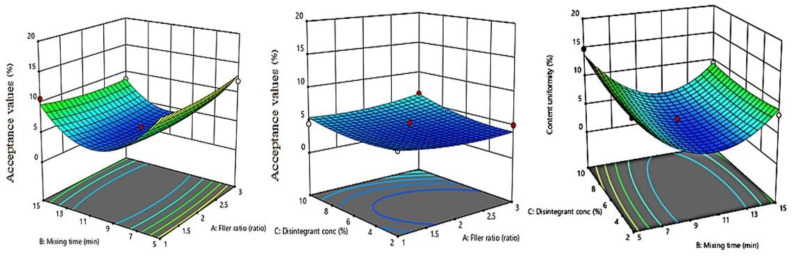
Influence of independent variables on the Acceptance value (Y4) with significance (**A**) *p* value = 0.9937, (**B**) *p* value = 0.1586, and (**C**) *p* value < 0.0134.

**Table 1 pharmaceuticals-15-01463-t001:** Flow characteristics of LX DT powder blends.

Formulation	Angle of Repose (°)	Carr’s Index (%)	Comment
F1	41	22	Passable
F2	42	23	Passable
F3	41	22	Passable
F4	42	21	Passable
F5	43	23	Passable
F6	41	21	Passable
F7	43	22	Passable
F8	44	22	Passable
F9	42	23	Passable
F10	43	23	Passable
F11	42	21	Passable
F12	44	22	Passable
F13	43	21	Passable
F14	42	23	Passable
F15	41	22	Passable

**Table 2 pharmaceuticals-15-01463-t002:** Evaluation of LX DT parameters.

FN	Hardness (kp)	Friability (%)	Thickness (mm)	Weight Variation (Average (mg) ± RSD%)	Dispersib Ility (s)	LX Content (%) ± SD	AV (%)
F1	2.45 ± 0.65	0.7	3.574	252.05 ± 3.50	11	101.41 ± 6.20	14.82
F2	4.06 ± 1.06	0.5	3.293	250.25 ± 3.07	7	99.65 ± 1.10	3.50
F3	3.72 ± 1.32	0.8	3.153	252.3 ± 2.10	13	102.46 ± 1.50	4.47
F4	4.3 ± 1.12	1	3.162	250.95 ± 2.66	12	103.45 ± 4.33	12.00
F5	2.87 ± 0.73	0.51	3.15	250.1 ± 2.22	18	101.20 ± 3.59	8.16
F6	4.16 ± 0.45	0.24	3.275	251.4 ± 2.70	6	98.80 ± 3.48	8.66
F7	3.62 ± 0.933	0.71	3.74	251.55 ± 2.96	20	101.77 ± 1.11	4.88
F8	2.85 ± 0.48	0.55	3.37	251.95 ± 4.44	8	98.81 ± 1.27	3.15
F9	4.13 ± 0.4	0.41	3.212	250.25 ± 2.20	8	101.29 ± 2.75	6.41
F10	2.52 ± 1.04	0.72	3.451	250.25 ± 3.43	12	98.60 ± 3.90	9.50
F11	4.09 ± 0.7	0.22	3.283	252 ± 3.09	6	96.08 ± 4.38	12.00
F12	4.84 ± 1.09	0.2	3.2	251.55 ± 3.19	16	101.12 ± 2.01	5.00
F13	4.31 ± 1.34	0.88	3.478	251.75 ± 2.95	13	101.44 ± 4.51	10.69
F14	3.82 ± 0.8	0.5	3.177	251.35 ± 2.96	18	98.54 ± 4.40	11.33
F15	3.00 ± 0.74	0.48	3.324	252.35 ± 3.47	7	96.54 ± 2.10	5.00

FN = Formulation number.

**Table 3 pharmaceuticals-15-01463-t003:** Design (Box–Behnken) of experiments with results.

Runs	Independent Variable			Dependent Variables							
				Observed Value				Predicted Value			
	X_1_	X_2_	X_3_	Y_1_	Y_2_	Y_3_	Y_4_	Y_1_	Y_2_	Y_3_	Y_4_
**1**	2	5	10	11.00	0.7	75.30	14.82	10.75	0.6588	75.79	**13.86**
**2**	2	10	6	7.00	0.5	79.30	3.50	7.33	0.5613	80.50	**3.88**
**3**	1	10	10	13.00	0.8	72.00	4.47	13.75	0.9401	70.76	**5.46**
**4**	1	5	6	12.00	1	75.80	12.00	11.50	0.8601	76.55	**11.97**
**5**	2	15	2	18.00	0.51	68.80	8.16	18.25	0.4638	68.31	**9.12**
**6**	3	15	6	6.00	0.24	83.00	8.66	6.50	0.2626	82.25	**8.69**
**7**	1	10	2	20.00	0.71	64.00	4.88	20.25	0.7626	64.21	**4.92**
**8**	2	10	6	8.00	0.55	83.10	3.15	7.33	0.5613	80.50	**3.88**
**9**	3	10	10	8.00	0.41	77.60	6.41	7.75	0.3601	77.39	**6.38**
**10**	2	15	10	12.00	0.72	73.90	9.50	11.75	0.6413	74.86	**9.50**
**11**	3	5	6	6.00	0.22	81.10	12.00	6.50	0.2801	80.83	**12.99**
**12**	3	10	2	16.00	0.2	65.40	5.00	15.25	0.1826	66.64	**4.01**
**13**	1	15	6	13.00	0.88	77.20	10.69	12.50	0.8426	77.48	**9.70**
**14**	2	5	2	18.00	0.5	66.00	11.33	18.25	0.4813	65.04	**11.33**
**15**	**2**	**10**	**6**	**7.00**	**0.48**	**79.10**	**5.00**	**7.33**	**0.5613**	**80.50**	**3.88**

X_1_ = MCC/Mannitol ratio (*w/w*); X_2_ = Mixing time (min); X_3_ = Disintegrant concentration (%); Y_1_ = Dispersibility time (sec); Y_2_ = Tablet friability (%); Y_3_ = Dissolution efficiency (%); Y_4_ = Content uniformity (Acceptance value%).

**Table 4 pharmaceuticals-15-01463-t004:** ANOVA analysis for the selected models for different responses of LXDTs.

Response	Model	Sequential *p*-Value	Lack of Fit *p*-Value	R^2^	Adjusted R²	Predicted R^2^	Adequate Precision	Significant Terms	PRESS	F Value
**Dispersibility time** **(Y1)**	Quadratic	0.0001	0.298	0.9897	0.9711	0.8650	21.1608	A, C, B^2^, C^2^	41.50	53.36
**Tablet friability** **(Y2)**	Linear	< 0.0001	0.1542	0.9119	0.8879	0.8242	18.2395	A, C	0.1419	37.95
**Dissolution efficiency (Y3)**	Quadratic	0.0124	0.8094	0.9432	0.8410	0.651	8.9001	C, C^2^	140.87	17.72
**Content uniformity (Y4)**	Quadratic	0.0011	0.3512	0.9586	0.8841	0.4805	9.8471	B, B²	96.65	12.87

A = Filler ratio. B = Mixing time. C= Disintegrant conc.

**Table 5 pharmaceuticals-15-01463-t005:** Composition and predicted, observed values of the responses for the optimized LX dispersible tablet formula.

Ingredient	Mg/Tablet	Response	Predicted	Observed ± SD
MCC	166.06			
Mannitol	55.35	Dispersibility time	5.17	4.4 ± 0.63
CP	15.58	Tablet friability	0.27	0.19
LX	8	Dissolution efficiency	81.92	80.64 ± 2.45
SLS	5	Content uniformity (AV)	4.37	4.65 ± 0.13

**Table 6 pharmaceuticals-15-01463-t006:** Quality control results of LXDTs during accelerated stability study. Data are represented as Mean ± SD (n = 3).

Parameter	Initial Time	1 Months	3 Months	6 Months
Hardness (kp)	4.01 ± 0.70	3.98 ± 0.85	4.0 ± 0.58	3.94 ± 0.75
Friability (%)	0.360	0.328	0.431	0.530
Thickness (mm)	3.426 ± 0.12	3.36 ± 0.134	3.40 ± 0.141	3.41 ± 0.113
Dispersibility time (s)	4	4	5	4
LX content %	99.62 ± 0.77	99.32 ± 0.72	98.05 ± 1.03	95.14 ±1.41

**Table 7 pharmaceuticals-15-01463-t007:** In vitro dissolution study at different times during accelerated stability study.

LXDT					Xefo ^®^ 8 mg
Initial Time		1 Months	3 Months	6 Months	
Time (min)	Mean ± SD	Mean ± SD	Mean ± SD	Mean ± SD	Mean ± SD
**5.0**	70.70 ± 0.90	69.07 ± 1.02	67.93 ± 0.68	66.87 ± 1.15	60.2 ± 2.10
**10.0**	80.30 ± 2.2	79.43 ± 1.17	78.12 ± 0.78	76.9 ± 1.32	70.51 ± 1.02
**20.0**	88.31 ± 2.4	87.37 ± 1.29	85.93 ± 0.86	84.60 ± 1.45	82.1 ± 1.36
**30.0**	92.73 ± 2.6	91.74 ± 1.36	90.22 ± 0.90	88.81 ± 1.52	89.71 ± 0.7
**60.0**	95.51 ± 2.62	95.41 ± 1.41	93.84 ± 0.94	92.37 ± 1.59	93.45 ± 2.4

**Table 8 pharmaceuticals-15-01463-t008:** Variables in Box–Behnken design for Lornoxicam DT.

Independent Variables, Factor	Low (−1)	Middle (0)	High (1)
**X1: MCC/ Mannitol ratio (*w*/*w*)**	1	2	3
**X2: Mixing time (min)**	5	10	15
**X3: Disintegrant concentration (%)**	2	6	10
**Dependent variables, Response**			
**Y1: Dispersibility time (sec)**			
**Y2: Tablet friability (%)**			
**Y3: Dissolution efficiency (%)**			
**Y4: Content uniformity (Acceptance value)**			

**Table 10 pharmaceuticals-15-01463-t010:** Composition of different formulae of Lornoxicam DT.

	Ingredients				
Formulation	MCC%	Mannitol%	CP%	LX%	SLS%
F1	56.53	28.27	10	3.2	2
F2	59.2	29.6	6	3.2	2
F3	42.4	42.4	10	3.2	2
F4	44.4	44.4	6	3.2	2
F5	61.87	30.93	2	3.2	2
F6	66.6	22.2	6	3.2	2
F7	46.4	46.4	2	3.2	2
F8	59.2	29.6	6	3.2	2
F9	63.6	21.2	10	3.2	2
F10	56.53	28.27	10	3.2	2
F11	66.6	22.2	6	3.2	2
F12	69.6	23.2	2	3.2	2
F13	44.4	44.4	6	3.2	2
F14	61.87	30.93	2	3.2	2
F15	59.2	29.6	6	3.2	2

## Data Availability

Not applicable.

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
