# Peer review of "Design and Optimization of Lornoxicam Dispersible Tablets Using Quality by Design (QbD) Approach"

_pharmaceuticals, 2022, doi:10.3390/ph15121463_

Round 1
Reviewer 1 Report
-Why is there no positive control (Lornoxicam tablets) in all experiments, especially the dissolution profile?
- Some figures should be improved to higher quality like figure 4
- In conclusion, the authors mention (as compared with the commercial tablet), even they used xefo tablets in stability studies for dissolution only.
Author Response
Manuscript ID: pharmaceuticals-1947316
Type of manuscript: Article
Title: Design and optimization of lornoxicam dispersible tablets using quality by design (QbD) approach
Reviewer’s comments:
Thank you for sending the reviewer’s comments regarding our submitted manuscript titled “Design and optimization of lornoxicam dispersible tablets using quality by design (QbD) approach”. We would like to thank the editor, and reviewers for their constructive comments.
We have carefully considered all the minor/major comments made by the editor/reviewer and revised the manuscript accordingly. The changes are made under “track change” format (a clean version of the manuscript is also provided)
Below we present our point by point responses to the reviewer/editorial comments
Reviewer 1:
Comments and Suggestions for Authors
-Why is there no positive control (Lornoxicam tablets) in all experiments, especially the dissolution profile?
Ans: We would like to thank the reviewer for the comments, There is a comparison with the optimized formula in dissolution profile
- Some figures should be improved to higher quality like figure 4
Ans: We would like to thank the reviewer for the comments. We have modified Figure 4 to improve the quality according to your suggestion.
- In conclusion, the authors mention (as compared with the commercial tablet), even they used xefo tablets in stability studies for dissolution only.
Ans: There is a comparison with the optimized formula in dissolution profile which showed similarity between the optimized formula with the innovator brand
Reviewer 2 Report
The manuscript entitled "Design and optimization of lornoxicam dispersible tablets using quality by design (QbD) approach" is quite interesting as a study. However, there is some incomplete information in the manuscript, including:
1. Was the SEM analysis done as stated in the abstract? but after searching there is no SEM data in the paper
2. Writing tables are not sequential.
3. Writing equations is also not in order
4. What is the essence of particle size testing?
5. If analyzing using DSC or FTIR, it is better if all single samples including excipients are also analyzed and compared to the mixed results in order to know the difference easily
6. The study of data or results is not comprehensive
7. Improved table display to make it more interesting
Author Response
Manuscript ID: pharmaceuticals-1947316
Type of manuscript: Article
Title: Design and optimization of lornoxicam dispersible tablets using quality by design (QbD) approach
Reviewer’s comments:
Thank you for sending the reviewer’s comments regarding our submitted manuscript titled “Design and optimization of lornoxicam dispersible tablets using quality by design (QbD) approach”. We would like to thank the editor, and reviewers for their constructive comments.
We have carefully considered all the minor/major comments made by the editor/reviewer, and revised the manuscript accordingly. The changes are made under “track change” format (a clean version of the manuscript is also provided)
Below we present our point by point responses to the reviewer/editorial comments
Reviewer 2:
Comments and Suggestions for Authors
The manuscript entitled "Design and optimization of lornoxicam dispersible tablets using quality by design (QbD) approach" is quite interesting as a study. However, there is some incomplete information in the manuscript, including:
- Was the SEM analysis done as stated in the abstract? but after searching there is no SEM data in the paper
Ans: SEM data were omitted from the manuscript to be concise
- Writing tables are not sequential.
Ans: Thank you for the comments. We have sequentially put them all now
- Writing equations is also not in order
Ans: We have corrected all the equations in orderly
- What is the essence of particle size testing?
Ans: To ensure the uniformity of particle size distribution and to ensure that no rapid sedimentation could occur after dispersion.
- If analyzing using DSC or FTIR, it is better if all single samples including excipients are also analyzed and compared to the mixed results in order to know the difference easily
Ans: We thank for the very valuable comments. We have analysed all the excipients and added in the revised manuscript.
- The study of data or results is not comprehensive
Ans: We have thoroughly revised data and modified substantially
- Improved table display to make it more interesting
Ans: We have modified the tables for better presentation
Round 2
Reviewer 1 Report
Accepted in the present form
Reviewer 2 Report
The paper has been revised according to the reviewer's expectations to make it more suitable for publication.